# Memory-Efficient LLM Training with Dynamic Sparsity: From Stability to Practical Scaling

Qiao Xiao [* 1]  Boqian Wu [* 2 3]  Patrik Okanovic [4]  Tomasz Sternal [4]  Maurice van Keulen [3]  Elena Mocanu [3]
Mykola Pechenizkiy [1]  Decebal Constantin Mocanu [2]  Torsten Hoefler [4]

## Abstract

Dynamic Sparse Training (DST) offers a promising paradigm for improving the training and inference efficiency of deep neural networks; however, we find that in large language model training, DST can suffer from optimization instability, manifested as loss spikes after topology updates. In this work, we show that the naive use of standard Adam-based optimizers leads to a cold-start issue for newly regrown parameters, resulting in excessively large updates and disrupted training dynamics. To address this issue, we propose Sparse Memory-Efficient Training (SMET), which stabilizes DST with optimizer warm-up and improves training progress through density-aware learning-rate scaling. SMET further reduces memory consumption by storing gradients and optimizer states only for active parameters. We provide a theoretical analysis of the update behaviors under SMET, showing improved optimization stability. Extensive experiments demonstrate that SMET enables stable, scalable, and memory-efficient sparse pretraining of LLMs, paving the way for sparse training as a practical alternative to dense training. Our code is publicly available at: https://github.com/QiaoXiao7282/SMET.

## 1. Introduction

The rapid scaling of deep learning models has driven remarkable progress across diverse tasks, yet it has simultaneously incurred prohibitive computational and memory costs. This tension has catalyzed extensive research into enhancing efficiency in both the training and inference (Zhuang et al., 2023; Ding et al., 2023; Zhou et al., 2024; Zhao et al., 2024;

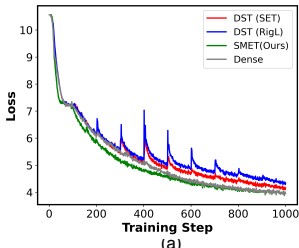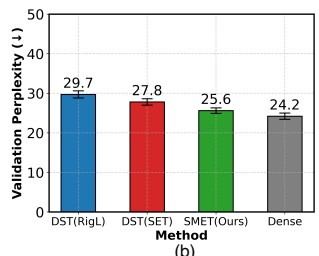

*Figure 1.* (a) Comparison of training curves between DST methods and dense training for LLaMA-240M on C4 dataset, with the topology update frequency set to every 100 steps for DST. (b) Validation perplexity (↓) comparison of dense training, SMET, and other DST methods at a density level of 0.25 on LLaMA-240M trained on the C4 dataset with 1.3B tokens.

Han et al., 2024; Okanovic et al., 2025).

Sparsity has emerged as a key technique to mitigate the growing computational costs of deep neural networks by retaining only a fraction of model weights (Hoefler et al., 2021; Frantar & Alistarh, 2023; Zhu et al., 2024). While traditional pruning typically targets post-training compression, another line of work explores weight sparsity from initialization to reduce computational overhead throughout training (Lee et al., 2019; Tanaka et al., 2020). Dynamic Sparse Training (DST), a prominent approach in this direction, allows the sparse topology to evolve through iterative connection pruning and regrowth (Mocanu et al., 2018; Evci et al., 2020). Prior work has demonstrated that this dynamic evolution allows the model to adaptively search for effective sparse connectivity guided by training dynamics. Consequently, DST can achieve competitive performance while maintaining high sparsity (Liu et al., 2021; Yuan et al., 2021; Lasby et al., 2024).

However, the application of DST to foundation models, particularly Large Language Models (LLMs), remains underexplored and faces several challenges. Current DST approaches largely inherit training recipes originally designed for dense models, which may be suboptimal in sparse regimes. For instance, learning rate schedules and initialization strategies optimized for dense training often fail to account for sparse settings, leading to mismatches as mod-

[1]Eindhoven University of Technology [2]University of Luxembourg [3]University of Twente [4]ETH Zürich. Correspondence to: Boqian Wu <boqian.wu@uni.lu>, Qiao Xiao <q.xiao@tue.nl>.

*Proceedings of the 43rd International Conference on Machine Learning*, Seoul, South Korea. PMLR 306, 2026. Copyright 2026 by the author(s).

els scale (Evci et al., 2022b; Nowak et al., 2024; Dey et al., 2024). Furthermore, the current hardware and software ecosystem provides limited acceleration support, especially for fine-grained sparsity, which restricts the practical efficiency gains of sparse training. These limitations have hindered the scaling of DST to large foundation models.

In this work, we show that the naive use of standard Adam-based optimizers (Kingma & Ba, 2015) struggles to support stable dynamic sparse training and often hinders training convergence. We empirically observe that the prune-and-regrow cycle induces pronounced loss spikes (see Figure 1(a)). Further analysis suggests that these spikes are primarily triggered by newly regrown connections, and this instability appears consistently across different DST methods, becoming more pronounced as model scales or topology updates become more aggressive. We attribute this behavior to a **"cold-start"** effect of newly regrown weights. Since these weights are activated without accumulated optimizer states, their early updates can be excessively large and poorly calibrated relative to mature weights. Such abrupt updates can disrupt the training trajectory, leading to transient loss spikes and degraded optimization stability.

To address these issues, we introduce an optimizer-state warm-up strategy that stabilizes the early updates of newly regrown weights and mitigates post-regrowth loss spikes, as shown in Figure 1 (a) (green curve). We further apply density-aware learning-rate scaling to compensate for the conservative updates introduced by warm-up and to account for the changed effective parameterization of sparse models. As a result, our method significantly improves over existing DST methods (see Figure 1 (b)).

Furthermore, to move beyond the limitations of binary masking in conventional DST implementations, we develop a memory-efficient sparse optimization approach that stores gradients and optimizer states only for active parameters. This makes the benefits of DST more practical for large-scale models such as LLMs. Unlike recent memory-saving methods that compress optimizer states during training but ultimately produce dense models (Zhao et al., 2024; Huang et al., 2025), our approach preserves sparsity throughout the entire training lifecycle, enabling potential efficiency benefits for inference after training.

Our contributions are summarized as follows:

1. We identify and analyze the loss spiking phenomenon in DST for LLMs pre-training, showing that newly regrown parameters can disrupt learning dynamics and degrade training stability after topology updates.

2. We propose a specialized DST method for LLMs pre-training that combines optimizer-state warm-up with density-aware learning-rate scaling to stabilize topol-

ogy updates and improve training performance.

3. We develop a memory-efficient sparse optimization implementation that stores gradients and optimizer states only for active parameters, supporting both unstructured and block-wise sparsity patterns while preserving sparsity throughout training.

4. We provide a systematic evaluation of DST for LLMs training across model scales and sparsity levels. Our results show that SMET significantly outperforms existing DST methods and can achieve marginal performance degradation at high sparsity on larger models.

## 2. Related Work

### 2.1. Memory-Efficient LLMs Training

Large language models (LLMs) incur substantial memory overhead during training. As a result, many prior works have explored techniques to reduce memory consumption (Zhao et al., 2024; Zheng et al., 2024; Huang et al., 2025). For instance, low-rank adaptation methods, such as LoRA and its variants, decompose weight updates into low-rank factors, significantly reducing the number of trainable parameters and the associated optimizer states (Hu et al., 2022; Dettmers et al., 2023; Lialin et al., 2024). Beyond low-rank parameterization, there are other approaches that focus on reducing the memory footprint of optimization by compressing (Park & Lee, 2025; Huang et al., 2025), or offloading (Robert et al., 2025) optimizer states, as well as modifying optimization dynamics to reduce auxiliary memory costs (Zhang et al., 2025). More recently, gradient low-rank projection methods restrict gradients to low-dimensional subspaces, lowering optimizer memory costs while largely preserving training dynamics (Zhao et al., 2024).

While these methods reduce memory cost to some extent, they typically produce dense models after training. In contrast, our work builds on sparsity to improve memory efficiency while maintaining stable optimization, preserving sparse models throughout training. This may also provide potential benefits for inference efficiency.

### 2.2. Dynamic Sparse Training

Dynamic Sparse Training (DST) (Mocanu et al., 2018; Evci et al., 2020; Chen et al., 2021; Yuan et al., 2021; Wu et al., 2025), as a sparse-to-sparse training paradigm, trains models from scratch while maintaining only a subset of parameters throughout training. By enforcing sparsity during both training and inference, DST offers a promising approach toward reducing both computational and memory costs while preserving model accuracy.

Prior studies have investigated the optimization dynamics and mechanisms of DST (Liu et al., 2021; Evci et al., 2022b).

Beyond these analyses, DST has also been widely explored across diverse tasks and domains, including computer vision (Liu et al., 2023; Wu et al., 2024), continual learning (Sokar et al., 2023), reinforcement learning (Graesser et al., 2022; Tan et al., 2023), and time-series analysis (Xiao et al., 2022). However, despite these advances, DST remains underexplored in the context of large language model (LLM) training. In this paper, we try to address key challenges in scaling sparse-to-sparse training to large, efficiency-critical language models.

# 3. Loss Spikes during LLM Training with DST

## 3.1. Preliminaries: Dynamic Sparse Training

Dynamic Sparse Training (DST) is a sparse-to-sparse training paradigm that maintains a sparse network throughout the entire training process. In contrast to static sparse training (Lee et al., 2019; Tanaka et al., 2020), which fixes the sparse structure after initialization, DST periodically updates the sparse topology to better align with the evolving learning dynamics. Specifically, every $\Delta T$ training steps, a fraction $r$ of active parameters is removed according to a pruning criterion, and the same number of inactive parameters is activated according to a regrowth criterion, thereby maintaining sparsity efficiency throughout training.

**Pruning process.** DST removes parameters deemed less important based on a pruning score function $s_{\text{prune}}(\theta)$. Parameters with the lowest scores are selected for removal:

$$\text{Drop} = \text{Arg TopK}\left(-s_{\text{prune}}(\theta), k\right), \quad (1)$$

where $\text{Arg TopK}(v, k)$ returns the indices of the top-$k$ elements of vector $v$. The pruning score can be defined using various criteria, such as weight information, accumulated updates, or optimizer statistics.

**Regrowth process.** To compensate for the removed parameters and explore the topological search space, DST activates the same number of new parameters based on a regrowth score function $s_{\text{grow}}(\theta, \nabla\theta)$:

$$\text{Grow} = \text{ArgTopK}\left(s_{\text{grow}}(\theta, \nabla\theta), k\right). \quad (2)$$

The regrowth score may depend on gradient information (e.g., RigL (Evci et al., 2020)), accumulated updates (e.g., Sparse Momentum (Dettmers & Zettlemoyer, 2019)), random exploration (e.g., SET (Mocanu et al., 2018)), or other heuristic signals.

Different DST methods differ mainly in their choices of pruning and regrowth criteria, as well as the frequency of topology updates $\Delta T$. Despite these differences, DST maintains only a subset of active parameters while allowing the sparse topology to evolve throughout training. While DST has achieved notable success in vision tasks, its application

to large-scale language model training remains relatively less well understood.

## 3.2. Observation: Loss Spikes and the Cold-Start Effect

In this section, we present an analysis of the learning behavior of DST in large language model (LLM) pre-training. We focus on two representative and widely used DST methods, SET (Mocanu et al., 2018) and RigL (Evci et al., 2020), which differ primarily in their regrowth strategies, random and gradient-based regrowth, respectively. In the experiments, we observe several phenomena that significantly impact optimization stability and convergence for DST in LLM training.

① **Topology updates in DST trigger loss spikes.** We consistently observe sharp, transient increases in the training loss immediately following parameter regrowth events across various DST methods, a phenomenon that does not occur in dense training (see Figure 1(a)). The severity of these spikes is influenced by the scale and timing of topology updates. In particular, higher regrown rates, where a larger fraction of parameters are removed and regrown at once, lead to stronger spikes. Similarly, lower regrowth frequency exacerbates spikes by concentrating regrowth into infrequent bursts (see Appendix D).

② **Loss spikes hinder training convergence.** The emergence of loss spikes introduces a characteristic non-smooth trajectory that stands in stark contrast to the monotonic decay observed in dense training. Each topology update event triggers a sharp increase in loss, followed by a gradual recovery. However, repeated update events therefore produce a sawtooth-like loss pattern with frequent transient disruptions. Over the course of training, these interruptions accumulate, resulting in slower overall progress and delayed stable progress toward the optimum, as shown in Figure 1.

③ **Newly regrown parameters are the primary cause of loss spikes.** Our ablation studies show that topology updates without regrowth process (i.e., pruning only) do not induce loss spikes (Figure 2(a), green line), as the remaining optimizer states evolve continuously without disruption. Furthermore, preserving the Adam optimizer states for regrown weights substantially reduces, or even eliminates, these spikes (Figure 2(a), blue line).

These results suggest that newly regrown connections are particularly vulnerable to unstable updates. Unlike long-trained weights, these "infant" connections lack the accumulated optimizer history necessary to calibrate update magnitudes. We refer to this phenomenon as the **"cold-start"** effect of newly introduced parameters. Notably, in standard DST settings, optimizer states are not available a priori for newly regrown weights. Addressing this instability is therefore essential for achieving stable DST at scale.

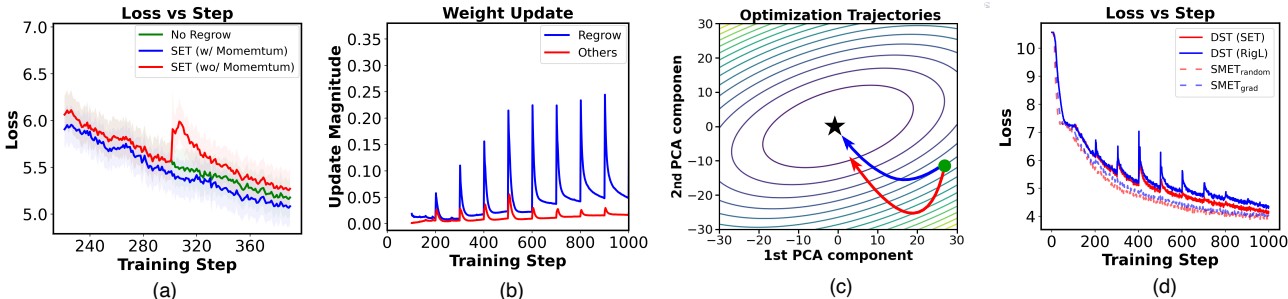

*Figure 2.* (a) Training curves for DST (e.g., SET) and its ablated variants from 200 to 400 steps, with a topology update occurring at step 300. (b) Comparison of weight update magnitudes between regrown and remaining weights with topology updates applied every 100 steps. (c) Illustrative training trajectories of SMET (blue) and other DST methods (e.g., SET) (red) immediately following a topology update. (d) Training curves comparing DST methods (RigL and SET) with SMET variants under different regrowth strategies. All experiments are conducted on LLaMA-240M trained on the C4 dataset under a density level of 0.25.

### 3.3. Understanding Loss Spikes in DST

In this section, we theoretically analyze the root cause of loss spikes in DST under Adam-based optimizer, which is widely used in LLM training, and examine how these spikes affect learning dynamics and hinder convergence.

In Adam, the magnitude of a parameter update is given by:

$$|\Delta\theta| = \alpha \cdot \frac{|\hat{m}_t|}{\sqrt{\hat{v}_t} + \varepsilon}. \tag{3}$$

where the optimizer states $\hat{m}_t$ and $\hat{v}_t$ accumulate gradient statistics over training iterations.

**Normal training behavior (mature weights).** For a weight that has been trained for many iterations, assuming that its local gradient distribution changes slowly at this stage of training, the bias-corrected moments approximate stable local first- and second-moment statistics: $\hat{m}_t \approx \mathbb{E}[g]$, $\hat{v}_t \approx \mathbb{E}[g^2]$. The effective per-parameter update magnitude can therefore be approximated as

$$|\Delta\theta_{\text{normal}}| \approx \alpha \cdot \frac{|\mathbb{E}[g]|}{\sqrt{\mathbb{E}[g^2]} + \varepsilon}. \tag{4}$$

For many mature parameters, gradient sign variations make the first-moment estimate smaller than the RMS gradient scale, i.e., $|\mathbb{E}[g]| < \sqrt{\mathbb{E}[g^2]}$. As a result, their effective update magnitude is typically below the nominal learning-rate scale $\alpha$.

**Training behavior of newly regrown weights.** In contrast, in DST the optimizer states $m$ and $v$ for newly regrown weights are necessarily initialized without prior moment information, since these parameters have not participated in any previous optimization steps. In Adam, when regrowth occurs late in training ($t \gg 1$), the bias correction terms effectively vanish ($1 - \beta_1^t \approx 1$ and $1 - \beta_2^t \approx 1$), leading to $\hat{m}_t \approx m_t$ and $\hat{v}_t \approx v_t$. Right after regrowth, the effective step size becomes:

$$|\Delta\theta_{\text{regrow}}| \approx \alpha \cdot \frac{1 - \beta_1}{\sqrt{1 - \beta_2}} \tag{5}$$

With the standard decay rates in Adam ($\beta_1 = 0.9, \beta_2 = 0.999$), this yields $|\Delta\theta_{\text{regrow}}| \approx 3.16\alpha$. Notably, this update magnitude is independent of the gradient scale and substantially exceeds the intended learning rate $\alpha$. A more detailed comparison between mature and newly regrown parameters under Adam is provided in Appendix A.

Consequently, mature parameters receive small, variance-normalized updates that are well aligned with the local curvature. In sharp contrast, newly regrown parameters experience disproportionately large and poorly calibrated updates due to the absence of historical optimizer statistics, as illustrated in Figure 2(b).

**Sudden disruption of training dynamics.** Regrown parameters introduce newly activated connections into the network. We hypothesize that large updates of newly regrown weights, especially when combined with a high learning rate, can abruptly alter the model's output, temporarily pushing it away from the previously established optimization equilibrium, as illustrated in Figure 2(c). Such abrupt changes lead to immediate and sharp increases in training loss, manifesting as spikes following regrowth events (see Appendix A for additional analysis). Repeated occurrences of such spikes produce a non-smooth, sawtooth-like training trajectory, which may adversely affect convergence.

## 4. SMET: Sparse Memory-Efficient Training

In this section, we introduce Sparse Memory-Efficient Training (SMET), a DST framework for stable and memory-efficient LLM training. SMET uses random regrowth by default, while its stabilization mechanisms can also be applied to other regrowth strategies. To address the training instability observed in DST for LLMs, SMET integrates optimizer-state warm-up to mitigate post-regrowth loss spikes, followed by density-aware learning-rate scaling to improve training performance under sparse updates. In addition, SMET stores sparse gradients and optimizer states only for active parameters, reducing the memory footprint during

LLM training. The overall training procedure is summarized in Appendix C.1, Algorithm 1.

## 4.1. Warm-up Strategies

Based on our analysis, in DST the loss spikes primarily arise from the "cold-start" of Adam optimizer moments, where newly regrown parameters lack accumulated historical statistics and thus receive excessively large updates. To address this issue, we introduce a warm-up strategy composed of the following two stages:

*First*, we propose to reset the timestep $t_i$ for each regrown parameter. This effectively reactivates Adam's internal bias correction mechanism for these connections:

$$\hat{m}_t = \frac{m_t}{1 - \beta_1^{t_i}}, \quad \hat{v}_t = \frac{v_t}{1 - \beta_2^{t_i}}. \tag{6}$$

For small $t_i$ (e.g., $t_i = 1$), $1 - \beta_1^{t_i} = 1 - \beta_1$ (small) and $1 - \beta_2^{t_i} = 1 - \beta_2$ (very small, e.g., 0.001 for $\beta_2 = 0.999$). Thus, in the first step:

$$v_t = (1 - \beta_2)g_t^2, \quad \hat{v}_t \approx \frac{(1 - \beta_2)g_t^2}{1 - \beta_2} = g_t^2, \tag{7}$$

This inflates the denominator such that $\sqrt{\hat{v}_t} + \epsilon \approx |g_t|$. Consequently, when the gradient scale does not change abruptly, the update magnitude follows the simplified bound (see Appendix B for the detailed derivation):

$$|\Delta\theta| \lesssim \alpha \cdot \sqrt{\frac{1 - \beta_2^{t_i}}{1 - \beta_2}}. \tag{8}$$

As $t_i$ increases, the update magnitude grows gradually as bias correction weakens. This process acts as an implicit warm-up, closely resembling the stable dynamics observed in early-stage training and effectively preventing update explosions for newly regrown parameters.

*Second*, complementary to the timestep reset, we introduce an explicit local linear warm-up for regrown parameters to further regulate update magnitudes. Specifically, the learning rate $\alpha$ is linearly increased from zero to the target $\alpha_s$ over $W$ steps following regrowth:

$$\alpha_t = \alpha_s \cdot \frac{t}{W}, \quad t = \{1, \ldots, W\},$$

where $W$ denotes the warm-up length. After the warm-up period ($t \geq W$), the parameter uses learning rate $\alpha_s$.

When combined, the implicit warm-up induced by timestep resetting and the explicit learning-rate warm-up act in a complementary manner, jointly preventing excessive updates of newly regrown parameters. This synergy effectively suppresses loss spikes and promotes stable optimization across different DST methods, as in Figure 2(d).

## 4.2. Compensation with Learning Rate Scaling

While warm-up strategies effectively neutralize loss spikes by constraining early post-regrowth updates, this conservative mechanism also temporarily reduces the effective update magnitude after topology changes. Meanwhile, sparsity can change the effective fan-in of sparse layers, thereby altering the appropriate learning-rate scale relative to dense training (Evci et al., 2022a; Dey et al., 2024). Together, these considerations motivate us to introduce a density-aware learning-rate scale-up.

Our scaling rule is motivated by Maximal Update Parameterization ($\mu$P) (Yang et al., 2021; Dey et al., 2024), which connects the appropriate learning-rate scale to the effective width or fan-in of a network. In sparse training, the active fan-in of a sparsified layer is reduced to approximately $d \cdot N_{\text{in}}$, where $d < 1$ denotes the parameter density and $N_{\text{in}}$ is the nominal input dimension (Evci et al., 2022a; Nowak et al., 2024). This reduction changes the effective parameterization relative to dense training, suggesting that the learning rate should be adjusted according to the density to maintain comparable update magnitudes in the sparse regime (Dey et al., 2024).

Accordingly, we scale the learning rate as: $\alpha_s = \alpha_0 \cdot \frac{1}{\sqrt{d}}$, where $\alpha_0$ is the base learning rate of dense training and $d$ denotes the current density. This scaling helps the update scale to account for the reduced effective fan-in of sparse layers, while also offsetting the conservative updates introduced by warm-up.

## 4.3. Memory-Efficient Optimization

For optimizers like Adam, maintaining two moment estimates ($m$ and $v$) per parameter introduces a memory overhead of approximately 2× the weights storage. However, conventional DST implementations typically rely on binary masks while still allocating full-sized optimizer states over the entire parameter space, thereby limiting the practical memory benefits of sparse training, especially for large-scale models.

To better realize the memory efficiency of sparse training, we store gradients and optimizer states only for active (non-zero) parameters, using an index-based representation. Concretely, for a sparse weight matrix $W \in \mathbb{R}^{n_{\text{out}} \times n_{\text{in}}}$ with density $d$, our implementation ensures that:

**Sparse Gradients.** We retain gradients only for active (non-zero) parameters by registering gradient hooks during backpropagation. Specifically, gradient entries are retained exclusively at the active indices of $W$, while gradients corresponding to inactive parameters are neither stored nor accumulated. Therefore, for sparse layers, gradient memory consumption is reduced by a factor of $d$ during training.

**Sparse Optimization States.** Similarly, the first and second moment estimates ($m$ and $v$) in Adam are stored only for the active parameters in sparse layers. Thus, the optimizer-state memory for these layers is proportional to the number of active parameters. For instance, when a layer is trained at density $d = 0.1$, its Adam state memory can be reduced substantially compared with storing full dense states.

**Index and Timestep Management.** Maintaining per-parameter timestep counters $t$ and active parameter indices introduces additional memory overhead. To reduce this cost, we adopt a block-wise storage strategy: parameters are grouped into blocks of size $B$ and active parameters within the same block share timestep and index metadata. This amortizes the storage cost of timestep and indices to $O\left(\frac{d \cdot n_{\text{out}} n_{\text{in}}}{B}\right)$ in sparse layers.

### 4.4. Local Stability of Regrowth Updates

In this section, we analyze how topology updates affect the local loss variation immediately after parameter regrowth. Assume the objective $\mathcal{L}(\theta)$ is $L$-smooth (i.e., $\nabla\mathcal{L}$ is $L$-Lipschitz). For any update $\Delta\theta_t = \theta_{t+1} - \theta_t$, the standard smoothness inequality holds:

$$\mathcal{L}(\theta_{t+1}) \leq \mathcal{L}(\theta_t) + \langle\nabla\mathcal{L}(\theta_t), \Delta\theta_t\rangle + \frac{L}{2}\|\Delta\theta_t\|_2^2. \quad (9)$$

Let $R_t$ denote the set of regrown parameters at step $t$, and let $\Delta\theta_t^{(R)}$ be the corresponding subvector of updates. In vanilla DST, newly regrown parameters may incur disproportionately large $\|\Delta\theta_t^{(R)}\|_2$, making the quadratic term $\frac{L}{2}\|\Delta\theta_t^{(R)}\|_2^2$ dominate and causing abrupt increases in loss immediately after regrowth.

In contrast, SMET constrains the update magnitude of regrown parameters during the warm-up phase. Specifically, SMET combines implicit warm-up induced by timestep resetting with an explicit learning-rate warm-up (Section 4.1). Together, these mechanisms yield the bound

$$\|\Delta\theta_t^{(R)}\|_2 \leq C_t, \quad (10)$$

where $C_t$ increases smoothly over the warm-up period, as detailed in Section 4.1. Substituting Eq. (10) into Eq. (9) shows that the local quadratic contribution from regrown parameters is bounded by $\frac{L}{2}C_t^2$. Moreover, the corresponding first-order term can also be controlled via Cauchy–Schwarz,

$$\left\langle\nabla\mathcal{L}(\theta_t), \Delta\theta_t^{(R)}\right\rangle \leq \|\nabla\mathcal{L}(\theta_t)\|_2 \cdot \|\Delta\theta_t^{(R)}\|_2$$
$$\leq \|\nabla\mathcal{L}(\theta_t)\|_2 \cdot C_t. \quad (11)$$

Since $C_t$ starts small and increases smoothly, SMET controls the local loss variation induced by newly regrown parameters immediately after regrowth, thereby suppressing

loss spikes and yielding a smoother optimization trajectory. This analysis focuses on the local effect of bounded regrowth updates rather than establishing a global convergence guarantee for Adam-based optimization.

## 5. Experiments

We evaluate SMET on large language model pre-training and study its effectiveness in terms of training performance, memory efficiency, and sparsity scalability.

**Architecture and hyperparameters.** We follow the experimental setup of (Zhao et al., 2024), which uses a LLaMA-based architecture with RMSNorm and SwiGLU activations (Zhang & Sennrich, 2019; Shazeer, 2020; Touvron et al., 2023). For each model size, we use the same set of hyperparameters across methods, except for the learning rate of DST-based methods. All experiments are conducted in BF16 format. Further details on experimental setups and hyperparameters are provided in Appendix C.

**Pre-training Dataset.** To evaluate the performance of SMET, we apply it to pre-train LLaMA-based language models on the C4 dataset. C4 is a cleaned version of the Common Crawl web corpus and is widely used for language model pre-training (Raffel et al., 2020). We further conduct additional experiments on the OpenWebText dataset (Gokaslan & Cohen, 2019) to evaluate the generality of SMET across different pre-training corpora.

**Baselines.** We compare SMET with representative memory-efficient training methods based on Adam-style optimization across different model sizes. **LoRA** (Hu et al., 2022) parameterizes weight updates with low-rank matrices, where only the low-rank factors are trained while the base weights remain frozen. **ReLoRA** (Lialin et al., 2024) extends LoRA to pre-training by periodically merging the low-rank updates into the main weights, followed by re-initializing the low-rank factors and resetting the optimizer states and learning rate. **GaLore** (Zhao et al., 2024) reduces optimizer memory by projecting gradients onto a low-dimensional subspace, enabling memory-efficient training while maintaining competitive performance.

*Table 1.* Perplexity comparison (↓) for pre-training LLaMA models of various sizes on the C4 dataset. Validation perplexity is reported. Results for baseline methods are obtained from (Zhao et al., 2024).

|  | **60M** | **130M** | **350M** | **1B** |
|---|---|---|---|---|
| Dense | 34.06 | 25.08 | 18.80 | 15.56 |
| LoRA | 34.99 | 33.92 | 25.58 | 19.21 |
| ReLoRA | 37.04 | 29.37 | 29.08 | 18.33 |
| GaLore | 34.88 | 25.36 | 18.95 | 15.64 |
| **SMET** | **33.38** | **25.21** | **18.90** | **15.61** |
| Training Tokens | 1.1B | 2.2B | 6.4B | 13.1B |

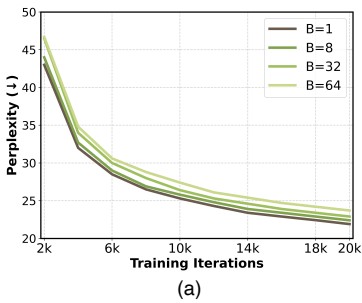 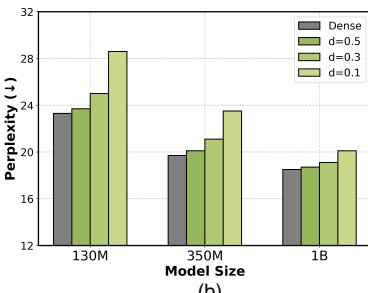 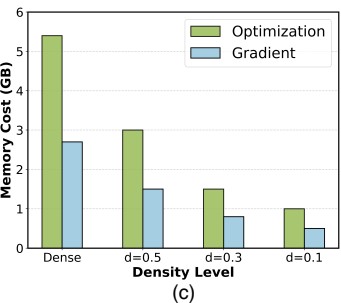

(a)                        (b)                        (c)

*Figure 3.* (a) Validation perplexity ($\downarrow$) curves for SMET under different block sizes $B$ for LLaMA-1B trained on the C4 dataset during 20k steps, under a density level of 0.1. (b) Validation perplexity ($\downarrow$) comparison between dense training and SMET across different density levels under different model sizes. (c) Memory cost ($\downarrow$) comparison of gradients and optimizer states between dense training and SMET across different density levels.

### 5.1. Performance Comparison

In this section, we evaluate the pre-training performance of SMET against representative memory-efficient baselines across different model sizes and training budgets.

Table 1 reports the validation perplexity of LLaMA models pre-trained on the C4 dataset at multiple scales. Across model sizes, SMET consistently outperforms other memory-efficient methods. Moreover, SMET achieves perplexity that is close to dense training, particularly at larger model scales. These demonstrate that SMET makes it a practical alternative to dense training for large language models.

We further evaluate SMET on a Qwen (Bai et al., 2023) model trained on the OpenWebText (Gokaslan & Cohen, 2019) dataset. The results, reported in Appendix F, show that SMET maintains consistent improvements beyond the LLaMA/C4 setting, suggesting that its benefits generalize across model architectures and pre-training corpora.

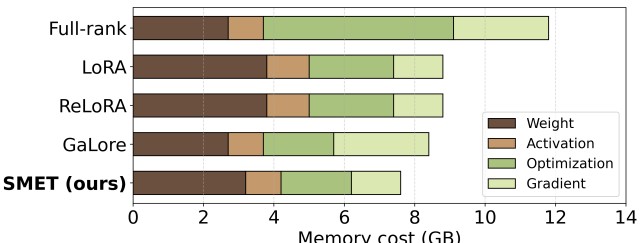

*Figure 4.* Estimated memory consumption for pre-training a LLaMA-1B model using different training methods in BF16.

### 5.2. Measurements of Memory Cost

In this section, we compare the memory consumption and its breakdown across different training methods. We analyze memory usage by component, including model weights, activations, gradients, and optimizer states. All measurements are conducted in BF16 with batch size 1 and gradient checkpointing disabled. The small batch size reduces the

influence of activation memory that scales with batch size, allowing the memory savings in gradients and optimizer states to be more clearly observed. Figure 4 presents the memory breakdown for the 1B-parameter model. We observe that SMET, at density $d = 0.4$, reduces the memory used by gradients and optimizer states. This confirms that storing gradients and optimizer states only for active parameters can translate into practical memory savings.

We further evaluate the trade-off between model performance and memory efficiency across different density levels. As shown in Figure 3 (b-c), reducing the density decreases the memory cost of gradients and optimizer states, while the perplexity gap remains modest for the 1B model. In particular, at density $d = 0.1$, SMET incurs only a small increase in perplexity while substantially reducing memory usage. These results highlight the potential of SMET for memory-efficient sparse training, especially in larger models where sparse training better preserves performance.

### 5.3. Evaluation on Block-Wise Sparsity

We further evaluate SMET under block-wise sparsity patterns. Specifically, we train LLaMA-1B on the C4 dataset with 5.2B tokens at density $d = 0.1$. During training, SMET maintains block-wise sparsity throughout the entire process, where both pruning and regrowth are performed at the block level rather than on individual weights. As shown in Figure 3(a), SMET remains stable across a wide range of block sizes. Even with a relatively large block size, e.g., $B = 32$, the validation perplexity increases by only about one point compared with the unstructured setting ($B = 1$). These results indicate that SMET is not limited to unstructured sparsity, but can also be naturally extended to structured block-wise sparsity.

In practice, block-wise sparsity is particularly appealing for practical deployment because it is more hardware-friendly than fully unstructured sparsity. Prior works have shown that block-structured sparsity can provide practical speedups

on modern hardware. By supporting block-wise sparse training, SMET preserves the potential for both memory savings during training and computational acceleration during inference. Using the block-wise sparsity implementation introduced in (Okanovic et al., 2025), we report additional inference-speed results in Appendix E, showing that the sparsity preserved by SMET can translate into practical inference speedups. This distinguishes SMET from memory-efficient training baselines that reduce training memory but ultimately produce dense models.

## 5.4. Effect on Sparsity Levels

We further study the effect of different sparsity levels on model performance and memory efficiency. Figure 3(b) shows the validation perplexity of different model sizes trained on the C4 dataset with 5.2B tokens. We find that higher sparsity generally results in higher perplexity, reflecting the expected trade-off between model capacity and efficiency. However, the performance gap between sparse and dense training becomes smaller as the model size increases. For the 130M model, lowering the density leads to a more visible degradation, while for the 1B model, sparse training remains much closer to dense training even at lower densities. This suggests that larger models are more tolerant to sparsity, making sparse training particularly attractive in the large-scale regime.

Figure 3(c) further demonstrates the memory benefit of sparse training. As sparsity increases, the memory required for both gradients and optimizer states decreases substantially, since SMET stores them only for active parameters. Therefore, higher sparsity not only reduces the number of trainable active parameters, but also directly lowers the memory footprint of gradient and optimizer-state storage.

## 5.5. Effect Across Regrowth Strategies

In this section, we evaluate SMET under different density levels and examine whether its benefits generalize across regrowth strategies. Table 2 reports validation perplexity at densities ranging from 0.5 to 0.1. We compare static sparse training with two vanilla DST baselines, SET and RigL, which use random and gradient-based regrowth, respectively. By default, SMET uses random regrowth for simplicity. We further evaluate a gradient-based variant, denoted as $\text{SMET}_{\text{grad}}$, which adopts the same regrowth rule as RigL while keeping the rest of the SMET training recipe unchanged.

As shown in Table 2, vanilla DST methods often underperform static sparse training, especially at lower densities. This aligns with the loss-spiking behavior observed after topology updates, which can disrupt optimization and slow convergence. In contrast, SMET consistently improves over SET across all density levels, while $\text{SMET}_{\text{grad}}$ similarly im-

*Table 2.* Validation perplexity (↓) under different density levels across regrowth strategies. Results are reported for LLaMA-350M trained on the C4 dataset with 2.6B tokens for 20k steps.

|  | d=0.5 | d=0.3 | d=0.2 | d=0.1 |
|---|---|---|---|---|
| Static Sparse | 22.90 | 23.55 | 25.40 | 27.44 |
| RigL | 25.75 | 27.83 | 28.85 | 29.90 |
| $\text{SMET}_{\text{grad}}$ | **22.52** | **24.01** | **24.78** | **26.50** |
| SET | 24.71 | 26.38 | 27.87 | 28.96 |
| SMET | **21.61** | **22.93** | **23.93** | **25.85** |

proves over RigL. These results indicate that the proposed stabilization strategy is not tied to a specific regrowth rule and can potentially benefit different DST methods.

## 5.6. Ablation Studies

We conduct ablation studies to better understand the contribution of each component in SMET. All ablation experiments are performed using LLaMA-350M trained on the C4 dataset with 2.6B tokens for 20k steps. We mainly examine the warm-up strategy, learning-rate scaling, and the topology update interval $\Delta T$.

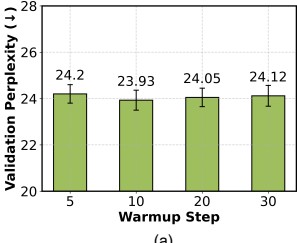 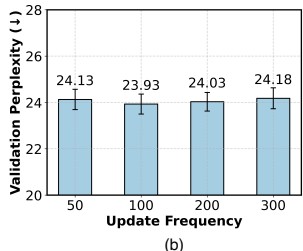

*Figure 5.* (a) Impact of warm-up steps $W$ on SMET. All experiments are conducted on LLaMA-350M trained on the C4 dataset under a density level of 0.2. (b) Ablation study on the topology update frequency $\Delta T$ in DST.

**Effect on warm-up and LR scaling.** We conduct ablation studies to analyze the contribution of the two key components in SMET: optimizer-state warm-up and density-aware learning-rate scaling. Table 3 reports the validation perplexity of LLaMA-350M on C4 under different density levels.

We observe that introducing warm-up improves training stability by smoothing early-stage optimization; however, the performance gains are limited when warm-up is applied in isolation. Applying learning-rate scaling further improves performance by adjusting the effective step size under sparse updates, but it remains insufficient to fully close the performance gap. When warm-up and learning-rate scaling are combined within SMET, the model consistently achieves the lowest perplexity across all density levels. In particular, we find that a warm-up length of $W = 10$ yields the

*Table 3.* Ablation of the key optimization components in SMET. Validation perplexity (↓) is reported for LLaMA-350M on C4 under different density levels. All configurations use Adam, and the full configuration corresponds to SMET.

|              | d=0.5     | d=0.3     | d=0.2     | d=0.1     |
|--------------|-----------|-----------|-----------|-----------|
| Adam         | 24.71     | 26.38     | 27.87     | 28.98     |
| + Warm-up    | 22.52     | 24.08     | 25.59     | 27.67     |
| + LR Scaling | 24.79     | 25.05     | 26.53     | 28.20     |
| SMET         | **21.61** | **22.93** | **23.93** | **25.85** |

best performance (see Figure 5(a)), striking a favorable balance between stability and optimization progress. Larger warm-up values do not provide additional benefits and may overly constrain early updates. These results demonstrate that the proposed components are complementary, and their integration in SMET yields the best overall performance.

**Effect on update frequency.** We study the impact of the update frequency in SMET by varying the update interval from 50 to 300 training steps, as shown in Figure 5(b). Our results show that an update frequency of 100 steps consistently achieves the best performance across different density levels. More frequent updates may introduce additional optimization noise, while less frequent updates degrade performance. Based on these observations, we adopt an update frequency of 100 steps as the default setting in all experiments unless otherwise specified.

## 6. Conclusion

We studied optimization instability in dynamic sparse training (DST) for large language models and identified the cold-start effect of newly regrown parameters as an important source of post-regrowth loss spikes. To address this challenge, we proposed Sparse Memory-Efficient Training (SMET), a stable and memory-efficient DST framework that combines optimizer-state warm-up for newly activated parameters with density-aware learning-rate scaling. SMET further reduces memory overhead by storing gradients and optimizer states only for active parameters, enabling more practical sparse training of large language models.

While SMET improves optimization stability and reduces the memory overhead of gradients and optimizer states, several challenges remain. First, our current design focuses on parameter sparsity, while activation memory during training and KV-cache memory during inference remain important sources of memory consumption. Second, translating sparsity into practical training speedups still depends on efficient hardware and software support, particularly for fine-grained sparse patterns. Future work includes extending SMET to activation sparsity, KV-cache compression, and developing more efficient sparse kernels for practical acceleration.

## Acknowledgements

This work was partly supported by H2020 SmartCHANGE, grant agreement No. 101080965 and TTW Perspectief MegaMind projects. This work was carried out partly using the Dutch national e-infrastructure with the support of the SURF Cooperative, using grant No. EINF-13990 and EINF-14276, and partly using the Luxembourg national supercomputer MeluXina. The authors gratefully acknowledge SURF and LuxProvide for their expert support.

## Impact Statement

This work focuses on improving the memory efficiency and stability of large language model training through optimization and sparsity techniques. By reducing memory overhead and improving training stability, our approach has the potential to lower the resource requirements for large models. We do not anticipate any broader social impacts beyond those commonly associated with machine learning research.

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

# Appendix

## A. Additional Analysis of Adam Cold-Start in DST

This section provides an expanded analysis of the Adam cold-start effect discussed in Section 3.3. We compare the effective update magnitudes of mature parameters and newly regrown parameters. The key observation is that newly regrown parameters have no accumulated optimizer history, while the global Adam timestep is already large, leading to poorly calibrated early updates.

**Normal late-stage training of mature parameters.**   For parameters that have been optimized for many iterations, and assuming that the local gradient distribution changes slowly, the bias-corrected moments approximate long-term gradient statistics:

$$\hat{m}_t \approx \mathbb{E}[g], \qquad \hat{v}_t \approx \mathbb{E}[g^2].$$

The corresponding effective update magnitude can be written as

$$|\Delta\theta_{\text{mature}}| \approx \alpha \cdot \frac{|\mathbb{E}[g]|}{\sqrt{\mathbb{E}[g^2]} + \epsilon}.$$

For many mature parameters, gradient sign variations make the first-moment estimate smaller than the RMS gradient scale. Therefore, their effective update magnitude is typically below the nominal learning-rate scale $\alpha$.

**Immediately after optimizer reset for regrown parameters.**   In contrast, for a newly regrown parameter whose optimizer states are initialized as $m = 0$ and $v = 0$, the first observed gradient $g_t$ gives

$$m_t = (1 - \beta_1)g_t, \qquad v_t = (1 - \beta_2)g_t^2.$$

When regrowth occurs late in training, the global Adam bias-correction factors are close to one, i.e.,

$$1 - \beta_1^t \approx 1, \qquad 1 - \beta_2^t \approx 1.$$

Thus,

$$\hat{m}_t \approx (1 - \beta_1)g_t, \qquad \hat{v}_t \approx (1 - \beta_2)g_t^2.$$

The corresponding update magnitude is

$$|\Delta\theta_{\text{regrow}}| \approx \alpha \cdot \frac{(1 - \beta_1)|g_t|}{\sqrt{(1 - \beta_2)g_t^2} + \epsilon}.$$

When the gradient magnitude is not dominated by $\epsilon$, this reduces to

$$|\Delta\theta_{\text{regrow}}| \approx \alpha \cdot \frac{1 - \beta_1}{\sqrt{1 - \beta_2}}.$$

With the standard Adam decay rates $\beta_1 = 0.9$ and $\beta_2 = 0.999$, this factor becomes

$$\frac{1 - \beta_1}{\sqrt{1 - \beta_2}} = \frac{0.1}{\sqrt{0.001}} \approx 3.16.$$

Therefore, newly regrown parameters can receive initial updates on the order of $3.16\alpha$, which is several times larger than the nominal learning-rate scale and can be much larger than the effective updates of mature parameters.

**Implications for loss spikes.**   Such abrupt increases in update magnitude can sharply increase $\|\Delta\theta\|$. Under a local second-order approximation,

$$\Delta\mathcal{L} \approx \nabla\mathcal{L}(\theta_t)^\top \Delta\theta + \frac{1}{2}\Delta\theta^\top H_t \Delta\theta,$$

the quadratic term grows with $\|\Delta\theta\|^2$ and may dominate the local descent term. This provides a plausible explanation for the transient loss increases observed immediately after regrowth. As the optimizer states of newly regrown parameters accumulate gradient history, their effective updates become better calibrated and the loss can gradually recover.

# B. Derivation of the Update Bound under Timestep Resetting

We provide additional analysis of how timestep resetting controls the early Adam updates of newly regrown parameters. This analysis complements the warm-up strategy described in Section 4.1. The key idea is that, after regrowth, Adam bias correction is computed using the local timestep $t_i$ of the newly regrown parameter rather than the global training step. This aligns the bias-correction factors with the actual optimization age of the parameter.

**Assumptions.** Consider a parameter that has just been regrown, with optimizer states initialized as $m_0 = 0$ and $v_0 = 0$. Let $t_i$ denote its local timestep after regrowth. We use standard Adam and omit the $\epsilon$ term for clarity; including $\epsilon$ only further reduces the update magnitude when the gradient is small.

**Bias-corrected moments with local timestep.** At local timestep $t_i$, the Adam moment estimates are

$$m_{t_i} = (1 - \beta_1) \sum_{k=1}^{t_i} \beta_1^{t_i - k} g_k,$$

and

$$v_{t_i} = (1 - \beta_2) \sum_{k=1}^{t_i} \beta_2^{t_i - k} g_k^2.$$

After bias correction, we have

$$\hat{m}_{t_i} = \frac{m_{t_i}}{1 - \beta_1^{t_i}}, \qquad \hat{v}_{t_i} = \frac{v_{t_i}}{1 - \beta_2^{t_i}}.$$

**Bounding the first moment.** Using the triangle inequality,

$$|m_{t_i}| \leq (1 - \beta_1) \sum_{k=1}^{t_i} \beta_1^{t_i - k} |g_k| \leq (1 - \beta_1^{t_i}) \max_{1 \leq k \leq t_i} |g_k|.$$

Therefore,

$$|\hat{m}_{t_i}| = \left| \frac{m_{t_i}}{1 - \beta_1^{t_i}} \right| \leq \max_{1 \leq k \leq t_i} |g_k|.$$

**Bounding the second moment.** For the second moment, retaining only the most recent gradient term gives

$$v_{t_i} \geq (1 - \beta_2) g_{t_i}^2.$$

After bias correction,

$$\hat{v}_{t_i} = \frac{v_{t_i}}{1 - \beta_2^{t_i}} \geq \frac{(1 - \beta_2) g_{t_i}^2}{1 - \beta_2^{t_i}},$$

which implies

$$\sqrt{\hat{v}_{t_i}} \geq |g_{t_i}| \sqrt{\frac{1 - \beta_2}{1 - \beta_2^{t_i}}}.$$

**Controlled update magnitude.** The Adam update magnitude satisfies

$$|\Delta \theta_{t_i}| = \alpha \frac{|\hat{m}_{t_i}|}{\sqrt{\hat{v}_{t_i}} + \epsilon} \leq \alpha \frac{|\hat{m}_{t_i}|}{\sqrt{\hat{v}_{t_i}}}.$$

Combining the above bounds yields

$$|\Delta \theta_{t_i}| \leq \alpha \cdot \rho_{t_i} \sqrt{\frac{1 - \beta_2^{t_i}}{1 - \beta_2}},$$

where

$$\rho_{t_i} = \frac{\max_{1 \leq k \leq t_i} |g_k|}{|g_{t_i}|}$$

captures the local variation of gradient magnitudes during the first $t_i$ steps after regrowth.

When the gradient scale does not change abruptly over these early local steps, $\rho_{t_i}$ remains close to one. In this common local regime, the update magnitude follows the simplified bound

$$|\Delta\theta_{t_i}| \lesssim \alpha\sqrt{\frac{1-\beta_2^{t_i}}{1-\beta_2}}.$$

This is the form used in the main text to describe the smooth growth of the effective update scale after timestep resetting.

**First-step behavior.** At the first local step $t_i = 1$, we have

$$m_1 = (1-\beta_1)g_1, \qquad v_1 = (1-\beta_2)g_1^2.$$

After bias correction,

$$\hat{m}_1 = g_1, \qquad \hat{v}_1 = g_1^2.$$

Therefore,

$$|\Delta\theta_1| = \alpha\frac{|g_1|}{|g_1|+\epsilon} \le \alpha.$$

Ignoring $\epsilon$, this corresponds to the standard Adam sign-step behavior with magnitude approximately $\alpha$. This contrasts with the global-step correction in vanilla DST, where a newly regrown parameter can receive an initial update of approximately

$$\alpha \cdot \frac{1-\beta_1}{\sqrt{1-\beta_2}} \approx 3.16\alpha$$

under the standard Adam setting $\beta_1 = 0.9$ and $\beta_2 = 0.999$.

**Early-step growth.** For early local steps and $\beta_2 \approx 1$, we have

$$1-\beta_2^{t_i} \approx t_i(1-\beta_2).$$

Thus,

$$\sqrt{\frac{1-\beta_2^{t_i}}{1-\beta_2}} \approx \sqrt{t_i}.$$

This shows that timestep resetting makes the update scale grow smoothly with the local timestep, instead of immediately jumping to the large global-step regime.

**Conclusion.** Timestep resetting controls the early updates of newly regrown parameters by aligning Adam bias correction with their local optimization age. At the first local step, the update magnitude is bounded by the nominal learning-rate scale $\alpha$, avoiding the large initial amplification caused by global-step correction. As $t_i$ increases, the update scale grows gradually. Combined with the explicit learning-rate warm-up in SMET, this provides a mechanism for introducing newly regrown parameters smoothly and mitigating post-regrowth loss spikes.

## C. Experimental Setup

In this section, we describe the LLaMA architectures and training hyperparameters used in our pre-training experiments. Table 4 summarizes the main architectural and training hyperparameters across different model sizes for SMET. Following the setup in (Zhao et al., 2024), all models use a maximum sequence length of 256 and a batch size of 512, corresponding to approximately 131K tokens per training step. For all experiments, we apply learning rate warm-up during the first 10% of the training steps, followed by a cosine annealing schedule that decays the learning rate to 10% of its initial value. All models are optimized based on Adam optimizer with $\beta_1 = 0.9$ and $\beta_2 = 0.999$.

For all model sizes ranging from 60M to 1B parameters, we observe that SMET remains stable under relatively larger learning rates compared to dense training and exhibits consistent behavior across scales. For dense baselines, we adopt learning rates $\{0.002, 0.001, 0.001, 0.001, 0.0005\}$ for increasing model sizes. For SMET, learning rates are determined using the proposed density-aware scaling strategy. Unless otherwise specified, we use a topology update frequency of 100 steps, an update ratio of 0.2, and a warm-up length of 10 steps for sparse training. We adopt a simple uniform density level across all layers during training, while keeping the embedding and output head layers dense.

*Table 4.* Hyperparameters of LLaMA models used in SMET pre-training. Data amounts are specified in tokens.

| Model | Hidden | Intermediate | Heads | Layers | Steps | Update Freq. | Update Ratio | Data Amount |
|-------|--------|--------------|-------|--------|-------|--------------|--------------|-------------|
| 60M   | 512    | 1376         | 8     | 8      | 10K   | 100          | 0.2          | 1.3B        |
| 130M  | 768    | 2048         | 12    | 12     | 20K   | 100          | 0.2          | 2.6B        |
| 240M  | 1024   | 2560         | 16    | 17     | 20K   | 100          | 0.2          | 2.6B        |
| 350M  | 1024   | 2736         | 16    | 24     | 50K   | 100          | 0.2          | 6.5B        |
| 1B    | 2048   | 5461         | 24    | 32     | 100K  | 100          | 0.2          | 13.1B       |

## C.1. SMET Algorithm

Algorithm 1 summarizes the overall training procedure of SMET. SMET follows the standard sparse-to-sparse training paradigm, where only a subset of parameters remains active during training and the sparse topology is periodically updated through pruning and regrowth. Compared with vanilla DST, SMET introduces three key modifications: optimizer-state warm-up for newly regrown parameters, density-aware learning-rate scaling, and sparse storage of gradients and optimizer states for active parameters only.

---

**Algorithm 1** Sparse Memory-Efficient Training (SMET)

---

**Require:** Training data $\mathcal{D}$, initial parameters $\theta_0$, density $d$, pruning ratio $r$, update interval $\Delta T$, warm-up length $W$, base learning rate $\alpha_0$

**Ensure:** Sparse parameters $\theta_T^{I_T}$ and active index set $I_T$

1: Initialize active index set $I_0$ with density $d$
2: Store only active gradient $g_0^{I_0}$ and optimizer states $m_0^{I_0}, v_0^{I_0}$
3: Set density-aware learning rate $\alpha \leftarrow \alpha_0/\sqrt{d}$
4: **for** training step $t = 0, \ldots, T - 1$ **do**
5:   Compute gradients $g_t^{I_t}$ and update only active parameters with sparse Adam
6:   **if** $(t + 1) \bmod \Delta T = 0$ **then**
7:     Prune indices $D_t \subset I_t$ according to $s_{\mathrm{prune}}$
8:     Regrow inactive indices $R_t$ randomly by default
9:     Update active set $I_{t+1} \leftarrow (I_t \setminus D_t) \cup R_t$
10:     Discard states for $D_t$, and warm up $R_t$ for $W$ steps
11:   **end if**
12: **end for**
13: **return** $\theta_T^{I_T}$ and $I_T$

---

In this work, random regrowth is used as the default choice for simplicity and efficiency. Other regrowth criteria, such as gradient-based regrowth, can be adopted by replacing the regrowth step while keeping the rest of the procedure unchanged. The sparse optimizer states are updated only for active parameters, and newly regrown parameters enter the optimization process through the warm-up schedule.

## D. Additional Analysis of Loss Spikes

This section provides additional evidence for the loss-spiking behavior discussed in Section 3.2. We analyze how the severity of loss spikes changes with topology update configurations and model scale. Overall, the results show that loss spikes become more pronounced when regrowth is more disruptive, either because more parameters are updated at once or because the model scale increases.

First, the severity of loss spikes is strongly influenced by the scale and timing of regrowth. A larger pruning ratio $r$ means that more active parameters are removed and regrown at each topology update, making the update more disruptive. Similarly, a longer update interval concentrates topology changes into less frequent but larger bursts. As shown in Figure 6(a), both larger pruning ratios and longer update intervals lead to sharper loss spikes, confirming that abrupt regrowth is a key factor behind the instability.

Second, loss spikes become more pronounced as model size increases. Figure 6(b) shows that, under the same DST configuration, larger models exhibit higher-amplitude spikes after topology updates. This suggests that the cold-start effect of newly regrown parameters can be amplified at larger scales, making naive DST increasingly difficult to stabilize for LLM training.

Together, these observations show that loss spikes are a consistent optimization challenge in conventional DST. Without mechanisms to control the early updates of newly regrown parameters, topology updates can repeatedly disrupt the training trajectory and limit the scalability of DST to large language models.

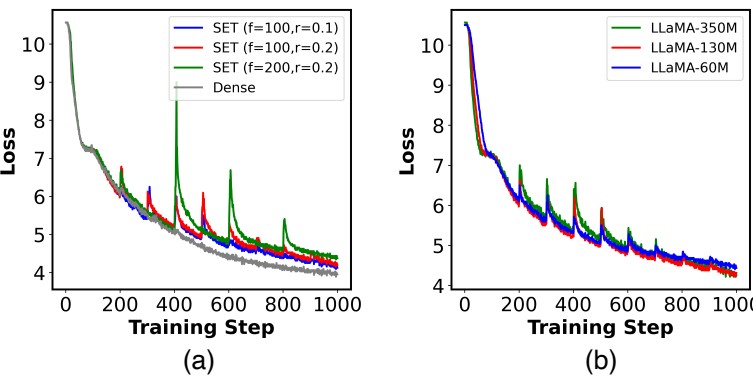

*Figure 6.* (a) Training curves comparing dense training with the SET method under different topology update frequencies and pruning ratios $r$ for LLaMA-240M on the C4 dataset. (b) Training curves of the SET method on the C4 dataset across different model sizes. All DST experiments are conducted at a density of 0.25.

## E. Inference Speed

In this work, although the main focus of SMET is to improve the stability and memory efficiency of dynamic sparse training, preserving sparsity after training can also provide potential inference-time benefits. Since SMET also supports block-wise sparsity, we measure the end-to-end inference speed of a sparse LLaMA-1B model under different sparsity levels and block sizes, using the method introduced in (Okanovic et al., 2025).

As shown in Figure 7, higher sparsity generally leads to larger inference speedups, while the block size also plays an important role in determining the practical acceleration. In particular, using block sparsity improves hardware efficiency compared with fully unstructured sparse patterns, making the sparse model more amenable to efficient inference. For example, at 70% sparsity, using a block size of 64 yields around $1.3\times$ speedup, and at 95% sparsity, the speedup increases to approximately $1.6\times$.

These results suggest that preserving sparsity throughout training enables SMET to provide practical inference benefits when combined with hardware-friendly sparse patterns. Nevertheless, the achieved speedup remains below the theoretical reduction in parameter count, indicating that sparse inference is still constrained by current kernel implementations and hardware support.

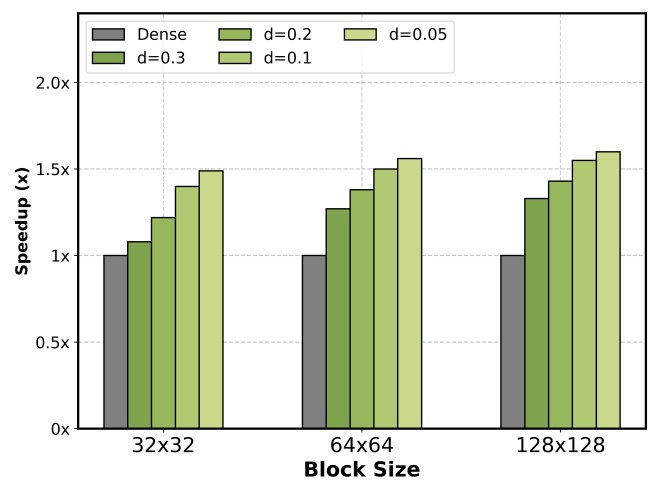

*Figure 7.* End-to-end inference speedup of LLaMA-1B under different block sizes and density levels.

## F. Evaluation on Additional Dataset

To further evaluate the generality of SMET beyond the C4 pre-training setting, we conduct additional experiments on the OpenWebText dataset. Specifically, we train a Qwen-0.6B model on 2.6B tokens from OpenWebText with a sequence length of 256 and a batch size of 256. We compare SMET with static sparse training and representative DST baselines under different density levels.

As shown in Table 5, SMET consistently improves over vanilla DST baselines across different density levels. Compared with RigL and SET, SMET achieves lower validation perplexity, indicating that the proposed optimizer-state warm-up and density-aware learning-rate scaling remain effective beyond the C4 dataset. The improvement becomes more pronounced at lower densities, where topology updates are more disruptive and stable optimization is more challenging.

These results further support the robustness of SMET across model architectures and datasets, suggesting that the proposed

stabilization strategy is not limited to a specific LLM family or pre-training corpus.

## G. Limitations and Discussion

While SMET improves the optimization stability and memory efficiency of dynamic sparse training (DST) for large language models, several limitations remain. Although SMET supports both unstructured and block-wise sparsity patterns, our empirical evaluation primarily focuses on unstructured sparsity and relatively simple block structures. Moreover, while SMET reduces memory overhead by storing optimizer states only for active parameters, exploring more hardware-aligned structured sparsity patterns and realizing practical acceleration remains an important direction for future work.

*Table 5*. Validation perplexity ($\downarrow$) on OpenWebText under different density levels. SMET uses random regrowth by default. Results are reported for Qwen-0.6B trained on 2.6B tokens.

|  | **d=0.5** | **d=0.25** | **d=0.125** |
|---|---|---|---|
| Static Sparse | 19.73 | 20.83 | 23.34 |
| RigL | 24.26 | 25.09 | 26.80 |
| SET | 22.83 | 23.40 | 24.23 |
| SMET | **18.60** | **19.84** | **21.80** |

In addition, our analysis and experiments focus on Adam-based optimizers, which are commonly used in large-scale LLM training. Whether similar cold-start effects arise under other adaptive or non-adaptive optimization schemes, and how SMET generalizes to such settings, warrants further investigation in future work.

Finally, while this work studies sparse pre-training, extending SMET to downstream fine-tuning, instruction tuning, and reinforcement learning from human feedback (RLHF) remains an interesting direction for future work. We believe that SMET provides a principled foundation for stabilizing dynamic sparsity at scale, and hope it will inspire further research at the intersection of optimization, sparsity, and efficient large-model training.

