# OpenReview forum: "Memory-Efficient LLM Training with Dynamic Sparsity: From Stability to Practical Scaling"
_ICML.cc/2026/Conference — ICML 2026 regular_

### Official Review · Reviewer_7G5y · 2026-03-10

**Soundness:** 2
**Presentation:** 1
**Significance:** 3
**Originality:** 3
**Overall Recommendation:** 4
**Confidence:** 3

**Summary:**

This paper proposes to enhance Dynamic Sparse Training (DST) stability by introducing a mechanism called SMET, which introduces the restarting and warm-up mechanisms to the regrown parameters. This has been motivated by the loss spikes observed in DST loss curves, which usually result from the regrowth updates. This paper also compares SMET with memory-efficient training baselines including GaLore, LoRA and ReLoRA, illustrating noticeable performance gains.

**Compliance With Llm Reviewing Policy:**

Affirmed.

**Final Justification:**

My recommendation is weak accept. The reason to suggest acceptance mainly derives from the impressive experimental results, which exhibit the efficiency of the proposed method.

**Key Questions For Authors:**

1. Can SMET be integrated to other memory-efficient algorithms?

2. What do you think is the main reason why SMET can beat dense baselines? If the results are solid, I believe it might be a promising direction to find the reason and help develop better dense training methods with improved performance.

**Limitations:**

yes

**Strengths And Weaknesses:**

Strengths:
1. The proposed method is simple yet effective, and its motivation is quite natural and easy to understand.
2. The practical performance is impressive.

Weaknesses:
1. I believe the presentation of the paper needs improvement. First of all, there are no algorithm tables throughout the paper, making it difficult to either understand the underlying DST method or the proposed SMET mechanism. Besides, the actual meaning of SMET is ambiguous. In Figure 2, SMET is regarded as an additional mechanism added to DST algorithms like SET or RigL, however, in Section 5, SMET is regarded as an independent algorithm, with no specific DST algorithms mentioned in the description. Such distinctions, together with the lack of an explicit algorithm table, have made it difficult for me to understand the experimental results completely.
2. Some experimental details are not clear enough. For example, in Table 1, what is the specific rank used for all baselines? Are they exactly the same as in Figure 4? If they are as reported in GaLore's original paper, their ranks are fewer than 1/4 of the hidden dimension, which do not appear to match the 0.4 density of SMET, from the perspective of memory consumption.
3. The experimental setups are somewhat strange. First of all, the compared baselines (LoRA, ReLoRA, and GaLore) are all low-rank training methods, not sparse training methods. Besides, the experiments only include pretraining models from 60M to 1B scales. Fine-tuning experiments are not conducted. The only task considered is pretraining on C4, with no downstream evaluation provided. The results in Table 1 show that SMET reaches a validation PPL of 18.90 when pretraining LLaMA-350M, however, SMET can only reach a PPL of 21.61 on the same task in Tables 2 and 3, even with a larger density of 0.5. Results for density 0.4 are also omitted in Tables 2 and 3, which is strange because density 0.4 has already been tested in other experiments.

---

> ### Author Rebuttal · Authors · 2026-03-30
>
> Thank you very much for your detailed comments and thoughtful questions.
> > w1: First of all, there are no algorithm tables throughout the paper….. Besides, the actual meaning of SMET is ambiguous…..
>
> We thank the reviewer for this helpful comment. SMET is not a new DST algorithm by itself; rather, it is a **wrapper built on top of an existing DST method (e.g., SET/RigL)**. In the current draft, we used “SMET” too loosely to refer to the overall sparse training pipeline, which blurred this distinction and made Figure 2 and Section 5 harder to interpret.
>
> We have revised this clearly by
>
> (i) adding an explicit algorithm pseudocode,
>
> (ii) separating the base DST method from the SMET mechanism, and
>
> (iii) clarifying section 5 and making it clear that we evaluate **DST+SMET**, rather than SMET as an independent topology-learning algorithm.
>
> > w2: Some experimental details are not clear enough…..
>
> We thank the reviewer for pointing this out. In Table 1, the baseline results follows the original settings provided in the GaLore paper.
>
> From the perspective of memory consumption, these baselines are reasonably comparable to our setting. In particular, under the reported GaLore configuration, the optimizer states **still account for more than half of the original optimizer memory, whereas our method operates at 0.4 density** and reduces the optimizer-state memory accordingly. In this sense, the comparison is intended to reflect a similar memory-efficiency regime.
>
> We will make the baseline configurations and their memory implications more explicit in the latest version.
>
> > w3: First of all, not sparse training methods. Besides, the experiments only include pretraining models from 60M to 1B scales. Fine-tuning experiments are not conducted, no downstream evaluation provided ….
>
> First, we compare against memory-efficient LLM training baselines because, to the best of our knowledge, our work is the first to scale DST to LLM pre-training from scratch.
>
> Second, we have added larger-scale and broader experiments: **SMET remains effective on larger model (e.g. LLaMA2-2B), on another model/data setting (Table), and in sparse fine-tuning on LLaMA2-7B Instruct with downstream evaluation.** These results strengthen the claim that the method is not limited to the original pre-training setup.
>
> Third, the discrepancy between Table 1 and Tables 2/3 **comes from different training budgets**. The ablation studies in Tables 2/3 are run for 10k steps (as noted in line 423), while Table 1 uses 60k steps in order to compare with the other baselines under their reported setting. We have made these settings more clearly in the latest version.
>
> Table 12. Sparse fine-tuning performance($\uparrow$) of LLaMA2-7B-Instruct on MMLU and GSM8K.
> |Model|MMLU | GSM8K
> |-|-|-|
> |wo/ SMET| 50.42 | 20.13
> |w/ SMET| 51.51 | 21.05
>
> Table 13. Perplexity($\downarrow$) for Qwen-0.6B trained on 2.6B openwebtxt dataset.
> |Model | dense | d=0.5 | d=0.25
> |-|-|-|-|
> |SET| 18.50| 22.69| 23.10
> |SET+SMET| 18.50 | 18.60 | 19.84
>
> Table 14. LM-eval performance($\uparrow$) of LLaMA2-2B after 2.6B training tokens.
> |Model|dense | d=0.5 | d=0.125
> |-|-|-|-|
> |SET| 36.81 |35.92 | 35.43
> |SET+SMET|36.81 |36.79 |36.23
>
> > w4: Can SMET be integrated to other memory-efficient algorithms?
>
> SPAM is a recent method for memory-efficient LLM training during dense training. We additionally combine SMET with SPAM in Table 1, and the results suggest that SMET is compatible with such methods and can **further improve their performance**.
>
> In particular, beyond optimizer-state savings, SMET can also provide gradient memory reduction, which is not directly addressed by SPAM.
>
> Table 15. Perplexity($\downarrow$) on LLaMA2-35M trained for 2.6B tokens.
> |Model|dense | d=0.5
> |-|-|-|
> |SPAM|22.10 | 23.74
> |SPAM+SMET|22.10 | 22.35
>
> > w5: What do you think is the main reason why SMET can beat dense baselines?...
>
> We thank the reviewer for this insightful comment. Our current hypothesis is that the gain comes from the interaction of **implicit regularization, dynamic topology exploration, and improved optimization stability**. Sparse training itself may provide a regularization effect, while DST can explore better sparse connectivity patterns during training. SMET then makes this exploration effective by stabilizing regrown parameters and preventing topology updates from causing destructive loss spikes.
>
> This interpretation is also consistent with our ablations: warm-up or learning-rate scaling alone brings only partial benefit, whereas their combination in SMET performs best.
>
> Thanks again for your time and effort. Please let us know if you still have any additional questions or would like further clarification.

---

> > ### Author Rebuttal · Reviewer_7G5y · 2026-04-04
> >
> > Thanks for the rebuttal. Most of my concerns are addressed, and I will maintain my positive rating.

---

> > > ### Author Response · Authors · 2026-04-05
> > >
> > > We sincerely appreciate your thorough review and valuable feedback. We are pleased that our response addressed your concerns, and your suggestions helped improve the quality of our paper. We also greatly appreciate your continued support and positive score.

---

### Official Review · Reviewer_k5Wo · 2026-03-11

**Soundness:** 2
**Presentation:** 3
**Significance:** 3
**Originality:** 2
**Overall Recommendation:** 4
**Confidence:** 4

**Summary:**

The paper addresses the problem of spiking loss after topology updates in dynamic sparse training (DST) by applying several changes to the optimization pipeline of DST. The proposed method (SMET) 1) resets the timestep for regrown parameters, decreasing the update magnitude for those parameters right after regrowth, implicitly adding a warm-up phase, 2) introduces an additional linear warm-up for the learning rate of regrown parameters, keeping the updates to newly regrown parameters small until the optimizer statistics have become meaningful, hence mitigating loss spikes after topology updates, and 3) scales the learning rate of each layer corresponding to its sparsity, leading to higher learning rates, especially for layers with high sparsity, reducing the impact of low-magnitude updates induced by the warm-up for newly regrown parameters. Additionally, efficient storing of sparse gradients and optimizer states reduces the memory cost during training. Experiments on LLaMA-2 models show that 1) SMET effectively mitigates loss spikes after parameter regrowth during DST, 2) SMET can train models at high sparsity levels without significant performance drops compared to dense training, and 3) SMET can outperform other memory-saving training methods in terms of validation perplexity of the final model while having a smaller memory footprint.

**Compliance With Llm Reviewing Policy:**

Affirmed.

**Final Justification:**

I decided to change my recommendation from 3 to 4 since all my concerns have been addressed:
- With the new stability analysis and the promise to rename the corresponding subsection to “Local Stability of Regrowth Updates”, my concerns about soundness are addressed.
- The memory breakdown in the authors' official comment addresses my concerns about Figure 4.
- Additional experiments in the rebuttal show that the method generalizes to different datasets and models.
- My concerns about actual memory reduction have been addressed with the promise to make the distinction between theoretical reduction and actual reduction more clear and with additional experiments showing the real reduction in memory footprint.

The proposed method, SMET, is a promising approach to sparse training, especially in the distributed setting as efficiently storing the sparse gradients reduces overhead from gradient synchronization.

I think 4 is the correct score for this paper as it is a significant contribution to sparse training literature, and in light of the rebuttal mostly technically sound. I decided to not give a score of 5 as I think the paper is lacking in originality. Bounding the change of regrown parameters is not far-fetched. Furthermore, the implementation of the update bound as well as the learning rate scaling are purely heuristic engineering choices.

**Key Questions For Authors:**

The 10x reduction in optimizer state memory cost at 10% density seems to be only theoretical. Can the authors provide benchmarks on memory savings for different densities?

**Limitations:**

yes

**Strengths And Weaknesses:**

### Strengths
- Overall, the paper is well written and easy to follow. DST and the problem of loss spikes are explained well and the motivation is clear.
- The method introduced to handle the Hessian approximation mismatch after regrowth is a solid contribution and, from what can be seen in the experimental section, works well in practice.
- The proposed method (SMET) addresses the challenge of large memory costs during training due to memory costs of optimizer states and gradients by sparsifying and storing them efficiently. Combined with methods for reducing activation memory cost, SMET could prove useful for distributed training as here activation memory might not outweigh the optimizer state and gradients as heavily. Additionally, storing the sparse gradients in an efficient structure could make gradient synchronization cheaper.
- Ablation studies analyze the different engineering choices that make up the proposed method and show that on their own they are not as effective. Overall SMET can improve the validation perplexity of DST trained models substantially for the architectures and dataset tested in the paper.


### Weaknesses
- Pruning itself is an orthogonal projection in parameter space and moves the model far away from the current location, making the optimizer statistics somewhat useless already. Additionally, regrowing parameters leads to parameters with wrong optimizer statistics, completely messing up the Hessian approximation used by Adam-variants to regularize descent directions. Though it is interesting that ablations show that pruning alone produces much smaller or even no loss spikes, it is somewhat obvious that wrong optimizer statistics lead to loss spikes. Hence, the contribution regarding the analysis of training loss after topology updates is incremental.
- The theoretical contribution (bound on loss increase after parameter regrowth) is trivial and oversold as "Convergence and Stability Analysis". What is shown in that subsection is that the difference in loss after an update during warm-up on regrown parameters is bounded via some increasing $C_t$. There is no analysis on how this bound interacts with convergence guarantees of Adam-based optimizers, neither is there a theoretical argument regarding optimization stability other than "updates to regrown parameters are bounded during warm-up".
- The memory cost comparison in Figure 4 is done in with a training setup that would not be used for training LLMs in practice and skews the results heavily into prioritizing memory savings in optimizer states and gradients. In an actual pre-training setting, the batch size would be greater than 1 and the average sequence length would also be much larger than 256, leading the activation memory to dominate. Hence, this figure is somewhat misleading.
- The experiments are not very extensive. Only a single dataset and model family. Only experiments up to 1B parameters. Claims of SMET enabling efficient sparse training of LLMs are not convincingly supported.

---

> ### Author Rebuttal · Authors · 2026-03-30
>
> Thank you very much for your detailed comments.
> > Q1: Pruning itself is an orthogonal projection … Hence, the contribution regarding the analysis of training loss is incremental.
>
> We agree that, in hindsight, such mismatched after topology updates can hurt optimization. However, in prior DST literature, especially in vision, prune-and-regrow is often beneficial because dynamic topology exploration can outperform a fixed sparse mask [1]. In contrast, we find this **standard recipe does not transfer smoothly to LLM pre-training**.
>
> Our motivation for analyzing the loss spikes is therefore twofold:
> (1) we isolate this failure mode empirically and theoretically in the LLM setting; and
> (2) we design a targeted solution, supported by the analysis, that stabilizes regrowth and makes DST more scalable for LLM pre-training.
>
> In this sense, our contribution is an **explanation of why a previously effective DST paradigm becomes unstable in LLM pre-training, and how to fix it**.
>
> [1] Evci, Utku, et al. Rigging the lottery: Making all tickets winners. ICML 2020.
>
> > Q2: The theoretical contribution ... is trivial and oversold as "Convergence and Stability Analysis"….
>
> We agree that the subsection is essentially a local one-step bound on the loss. It does not analyze interaction with Adam’s convergence guarantees.
> In the revision, we will rename the subsection to **“Local Stability of Regrowth Updates”** and clearly frame it as a narrow local analysis aimed at mitigating the cold-start effect of newly regrown parameters.
>
> To make the contribution clearer, we will also add the stronger optimizer-level perspective the reviewer implicitly expects. Specifically, we will decompose each Adam update as
> $$
> \Delta \theta_t=\Delta \theta_t^{(M)}+\Delta \theta_t^{(R)}
> $$
> where superscripts M and R denote mature and regrown components. Substituting into the smoothness inequality yields
> $$
> \mathcal{L}(\theta_{t+1}) \leq \mathcal{L}(\theta_t)+\langle\nabla \mathcal{L}(\theta_t), \Delta \theta_t^{(M)}\rangle+\frac{L}{2}||\Delta \theta_t^{(M)}\||_2^2+\mathcal{E}_t^{\mathrm{reg}}
> $$
> with the residual bounded (under SMET) by
> $$
> \mathcal{E}_t^{\mathrm{reg}} \lesssim \|\nabla \mathcal{L} (\theta_t) \|_2 C_t+\frac{L}{2} C_t^2+L \|\Delta \theta_t^{(M)} \|_2 C_t .
> $$
> This shows that SMET primarily suppresses the extra perturbation from topology updates while preserving the dynamics of mature parameters.
>
> > Q3: The memory cost comparison …. Hence, this figure is somewhat misleading.
>
> We agree that in longer-context regimes, activation memory can indeed dominate. However, we would like to clarify that this setting is not entirely detached from practice. **In large-scale distributed pre-training**, the per-device micro-batch is often small, and activation memory is frequently managed with activation checkpointing. Similar component-wise presentations have also been used in prior work [2].
>
> More importantly, the central contribution is to **enable stable and scaling DST for LLM pre-training**. We believe that activation-memory optimizations can in principle be combined with SMET for further end-to-end savings, but somehow beyond the scope of the current paper.
>
> [2] Zhao, Jiawei, et al. Galore: Memory-efficient llm training by gradient low-rank projection. ICML 2024.
>
> > Q4: Only a single dataset and model family. Only experiments up to 1B parameters….
>
> To address this concern, we conducted additional experiments on other model families and datasets. As shown in Table 1, **SMET also works well on Qwen-0.6B trained on 2.6B tokens of OpenWebText. We further scale SMET to LLaMA2-2B**.
> These additional results suggest that the benefit of SMET is not limited to a single dataset or model family, and can scale beyond the 1B regime.
>
> Table 9. Perplexity($\downarrow$) for Qwen-0.6B trained on 2.6B openwebtxt dataset.
> |Model | dense | d=0.5 | d=0.25
> |-|-|-|-|
> |SET| 18.50| 22.69| 23.10
> |SET+SMET| 18.50 | 18.60 | 19.84
>
> Table 10. Perplexity($\downarrow$) on LLaMA2-2B trained for 2.6B tokens.
> |Model|dense | d=0.25 | d=0.125
> |-|-|-|-|
> |SET| 19.25 | 21.87 | 23.49
> |SET+SMET| 19.25 | 19.52 | 20.53
>
> > Q5: The 10x reduction ... seems to be only theoretical….
>
> We agree that the reduction is an idealized upper bound. In practice, the measured saving is smaller because not all layers are sparsified in our current implementation (e.g., the embedding layer). We will clarify this point in our latest version.
>
> More broadly, a more uniform sparsification of all components could further increase the practical memory saving.
>
> Table 11. Memory($\downarrow$) for gradient and optimizer states (GB) for LLaMA2-1B under different density level.
> |Model|dense | d=0.2 | d=0.1
> |-|-|-|-|
> |Gradient| 2.7| 0.8 | 0.5|
> |Optimzer| 5.4| 1.5 | 1.0|
>
> Thanks again for your time and effort. We hope the clarifications address your concerns and would appreciate reconsideration of the score.

---

> > ### Author Rebuttal · Reviewer_k5Wo · 2026-04-01
> >
> > I thank authors for their detailed response. I appreciate the response to Q3, however, I am still unsure how the memory savings in gradients and optimizer states compare to the activation memory. Could the authors provide a breakdown of memory allocation similar to Figure 4 for a realistic training setting, e.g. realistic micro batch size and sequence length, and checkpointing per transformer block?

---

> > > ### Author Response · Authors · 2026-04-01
> > >
> > > We thank the reviewer for this follow-up question. To address this more directly, we additionally measured the memory breakdown with activation checkpointing enabled and summarize the results below.
> > >
> > > These results show that, as expected, activation memory increases with sequence length and batch size. However, in this regime, the gradient and optimizer-state memory reduced by SMET remains substantial and non-negligible. In particular, under d=0.1, gradients and optimizer states together account for about 1.5 GB, whereas in **the dense case they would require roughly 2.7 + 5.4 GB**.
> > >
> > > Therefore, our point is not that optimizer/gradient memory always dominates activation memory, but that **SMET reduces an important memory component during training, while also enabling stable sparse training and improving performance.**
> > >
> > >
> > > Table 1. Memory($\downarrow$) comparision per device for LLaMA2-1B under density level of 0.1.
> > > |Model|Weight | Activation | Gradient | Optimizer
> > > |-|-|-|-|-|
> > > |bs=4; seq_len=256| 2.7| 0.3 | 0.5| 1.0
> > > |bs=4; seq_len=512| 2.7| 0.5 | 0.5| 1.0
> > > |bs=4; seq_len=1024| 2.7| 1.5 | 0.5| 1.0
> > > |bs=8; seq_len=1024| 2.7| 2.1 | 0.5| 1.0
> > >
> > > Thank you again for your time and effort. If you feel that our response has addressed your concern, we would greatly appreciate your reconsideration of the score.

---

### Official Review · Reviewer_Bmtc · 2026-03-11

**Soundness:** 3
**Presentation:** 3
**Significance:** 3
**Originality:** 3
**Overall Recommendation:** 4
**Confidence:** 4

**Summary:**

This paper identifies a cold-start inability in DST for LLMs where newly regrown parameters lack optimizer history and end up causing loss spikes. The observation is novel and interesting, they propose a new method SMET to mitigate this issue.

**Compliance With Llm Reviewing Policy:**

Affirmed.

**Final Justification:**

I maintain my score of 4. The paper is a meaningful contribution to dynamic sparse training, and the rebuttal addressed several concerns with helpful experiments.

In agreement with a fellow reviewer, I believe 4 is the correct score. The theoretical analysis remains my primary concern: it does not provide insight specific to SMET beyond what generic bounded-update arguments would yield. The authors clarify that SMET targets only regrown parameters unlike global clipping, but this is a design choice, not a theoretical insight; the analysis never formalizes why regrowth-specific intervention is provably better than global damping. Additionally, the proposed solutions follow naturally from the diagnostic insight, limiting the overall technical contribution.

**Key Questions For Authors:**

1. How does SMET perform at higher and more practical sequence lengths (2048+)? Right now, I think 256 is used? Are there any challenges to larger lengths?

**Limitations:**

Yes, the authors discuss it.

**Strengths And Weaknesses:**

Strengths:

1. Quite well motivated introduction and problem setup. The contributions 1 & 2 are also clear as opposed to prior work.
2. The idea authors propose is also quite interesting including the set of observations in section 3.2. All the claims here are well supported with thorough experimental validation. Particularly the ablation showing pruning-only causes no spikes while preserving optimizer states eliminates them (Fig. 2a). The cold-start analysis (Eq. 5) cleanly identifies the root cause.
3. The proposed method SMET is practical, seems easy to implement and requires no architectural changes--making it easier to adopt. The authors also provide a density-aware LR scaling to mitigate slow convergence  as a side-effect of linear warmup. They also provide memory-efficient implementation of their method which is crucial for at scale models.
4. The ablation studies are well-structured demonstrating that warmup and LR scaling are individually insufficient but complementary. The experimental validation (with performance-memory tradeoff) with integration results across SET and RigL confirms that method generalized across DST methods.


Weakness:
1. I am not sure on how novel is the memory-efficient optimization component while the motivation is sound and important - the method itself is not novel. If it is, I think authors should highlight the difference. It will make for a stronger argument towards SMET or is this component flexible and one can use some other memory-efficient techniques. The claim to novelty here is weak.
2. Additionally, they authors do not provide any analysis of training or computational speedups. Does the training speed improve or remain the same. In the paper, they say it has potential for computation speedups, but how remains uncertain.
3. The density-aware scaling - not sure how to validate if $1/\sqrt{d}$ is the right scaling? The authors use $\mu$P analogy but is change to the exponent of $d$ non-trivial? How to understand the behavior with alternative exponents like $d$ or $d^{0.75}$?
4. Convergence and stability analysis: the analysis assume global L-smoothness which is known to be a poor fit for transformers loss landscapes where curvature varies dramatically across parameter space. Moreover, even if this was true, the analysis only shows that bounded updates yield bounded loss increase -- which is a trivially true statement and provide no insight specific to SMET. You could achieve the exact argument with gradient clipping, multiplying by a small constant or smaller learning rate etc.
5. The authors don't compare with some latest work in the field e.g SPAM (the authors did cite it as well) -- not sure why this sort of  comparison is not made.


Minor comments:
1. Pruning criteria (line 132-141): authors should specify notation of $\theta$ for model parameter.

---

> ### Author Rebuttal · Authors · 2026-03-30
>
> Thank you very much for your detailed comments.
>
> > w1: I am not sure on how novel is the memory-efficient optimization …. The claim to novelty here is weak.
>
> We thank the reviewer for raising this concern. Our main novelty is not the memory-saving component alone, but the overall framework for **scaling DST in LLM pre-training**.  Specifically,
>
> (1) we identify a previously underexplored bottleneck in this setting, namely the cold-start instability of newly regrown parameters;
>
> (2) we propose SMET as a simple, pluggable framework that can be integrated with existing DST methods to address this issue; and
>
> (3) we show that this leads to stable and memory-efficient sparse pre-training at scale.
>
> > w2: Additionally, they authors do not provide any analysis of training or computational speedups….
>
> We agree that the submitted paper does not make the speed aspect sufficiently explicit, and we will clarify this in the revised version.
>
> For inference, since SMET naturally supports block-wise sparsity, which is more compatible with current hardware. **We observe up to 2× speedup at 90% sparsity (Table 4)**.
>
> For training, the current implementation is still slightly slower than dense training due to the overhead of index management and sparse element retrieval, despite the reduced number of operations.
>
> So the current benefit is clearer on inference, while training speedup remains a systems challenge for future work.
>
> Table 4. Inference speedup of LLaMA2-1B.
> |Sparsity|Latency(ms)|Throughput (TFLOPs)|
> |-|-|-|
> |0.00|256.94|236.41|
> |0.50|219.72|276.45|
> |0.70|182.07|333.63|
> |0.90|133.57|454.77|
> |0.95|120.00|506.18|
>
> Table 5. Running Time per Iteration. Measured by the average of 100 iterations under H100 GPU.
> |Method|LLaMA2-240m | LLaMA2-350m
> |-|-|-|
> |Adam| 0.724s | 0.826s
> |SMET| 0.757s | 0.847s
>
> > w3: How to validate if 1/sqrt(d) is the right scaling? … with alternative exponents like d or d^0.75?
>
> We thank the reviewer for this important question. The 1/sqrt(d) scaling is not chosen ad hoc. It is motivated by the theory in [1], as explained in section 4.2.
>
> In addition, we also conducted experiments with alternative scaling choices. These results show that **1/sqrt(d) works better in our setting**. We will clarify this choice is supported by both the theoretical justification in [1] and our empirical sensitivity analysis.
>
> Table 6. Perplexity($\downarrow$) on LLaMA2-350M trained for 2.6B tokens.
> |Model| 1/sqrt(d) | d | d^0.75
> |-|-|-|-|
> |d=0.5|21.61 |21.82 | 22.76
> |d=0.3| 22.93 | 23.21 | 24.56
>
> [1] Dey, Nolan, et al. Sparse maximal update parameterization: A holistic approach to sparse training dynamics. NeurIPS 2024.
>
> > w4: Convergence and stability analysis…. You could achieve the exact argument with gradient clipping, multiplying by a small constant or smaller learning rate etc.
>
> Our main point is not simply that “bounded updates imply bounded loss increase.” In DST, the instability is regrowth-specific: it is triggered by topology updates, concentrated on newly regrown parameters, and only appears during the short post-regrowth phase.
>
> This is why generic methods such as global clipping or a smaller global learning rate are not the right match, **they damp all parameters**, including mature ones that are already well behaved. By contrast, SMET acts only on the unstable subset and only when the mismatch occurs.
>
> > w5: The authors don't compare with some latest work in the field e.g SPAM ….
>
> SPAM and SMET target different settings. SPAM is mainly designed for dense training, while SMET is designed for dynamic sparse training, where the instability comes specifically from topology updates and newly regrown parameters.
>
> That said, SMET is compatible with SPAM. As shown in Table 1, combining SPAM + SEFT improves sparse training performance over SPAM alone. In addition, SMET can also reduce gradient memory, which is not directly addressed by SPAM.
>
> Table 7. Perplexity($\downarrow$) on LLaMA2-35M trained for 2.6B.
> |Model|dense | d=0.5
> |-|-|-|
> |SPAM|22.10 | 23.74
> |SPAM+SEFT|22.10 | 22.35
>
> > w6: How does SMET perform at higher and more practical sequence lengths (2048+)….
>
> To further examine practicality, we additionally evaluate SMET at longer sequence lengths up to 2048. The results show that **SMET remains effective at larger sequence lengths**. This suggests that the proposed stabilization mechanism is not limited to short-context training.
>
> The main challenge at longer sequence lengths is system-level memory pressure, especially from increased activation memory, rather than a limitation of SMET itself.
>
> Table 8. Perplexity($\downarrow$) on LLaMA2-350M under different sequence length, with bs=256.
> |Model|seq_len=256 | seq_len=1024 | seq_len=2048
> |-|-|-|-|
> |d=0.2|26.50 | 25.70 | 24.43
>
> We have revised these details in the new version. Please let us know if any point still needs clarification. Thanks again for your time and effort.

---

> > ### Author Rebuttal · Reviewer_Bmtc · 2026-04-03
> >
> > I thank the authors for their detailed response. I do not have any further questions and decide to maintain my score.

---

> > > ### Author Response · Authors · 2026-04-05
> > >
> > > Thank you for your positive rating and thoughtful feedback on our paper. We are pleased that our response addressed your concerns, and we greatly appreciate your time and insightful comments.
> > >
> > > Best regards,
> > >
> > > Authors

---

### Official Review · Reviewer_kp99 · 2026-03-20

**Soundness:** 3
**Presentation:** 3
**Significance:** 3
**Originality:** 3
**Overall Recommendation:** 4
**Confidence:** 3

**Summary:**

The authors propose Sparse Memory-Efficient Training (SMET), which stabilizes training using an optimizer warm-up for regrown parameters and density-aware learning-rate scaling. SMET also reduces memory overhead by storing optimizer states exclusively for active parameters. Experiments validate SMET by pre-training LLaMA models (up to 1 billion parameters) on the C4 dataset.

**Compliance With Llm Reviewing Policy:**

Affirmed.

**Key Questions For Authors:**

1. How might this method perform with optimization schemes e.g. muon, given the current focus on Adam-based optimizers?
2. Does the performance gap with dense training widen or narrow when scaling beyond the 1-billion parameter models tested?

**Limitations:**

yes

**Strengths And Weaknesses:**

### **Strengths**

*  This paper presents the first systematic evaluation of Dynamic Sparse Training (DST) for Large Language Model (LLM) pre-training. It thoroughly tests the method across multiple model scales (up to 1 billion parameters) and varying sparsity levels , while successfully evaluating both unstructured and structured (block-wise) sparsity settings.
* The authors provide strong empirical and theoretical analysis to identify the "cold-start" effect, showing that newly regrown parameters lack historical optimizer states and thus cause disruptive loss spikes. The proposed Sparse Memory-Efficient Training (SMET) method solves this by combining an explicit optimizer momentum warm-up with density-aware learning-rate scaling.
*  The work is grounded in theoretical analysis, demonstrating that SMET effectively bounds the update magnitudes for newly activated parameters during the warm-up phase. This bounded update tightly controls the per-step loss increase, suppressing spikes and yielding a smoother optimization trajectory.
*  By implementing an index-based representation that strictly stores gradients and optimizer states only for active (non-zero) parameters, SMET unlocks true memory efficiency. At a density level of 0.4, it reduces the overall memory footprint by nearly half compared to standard dense training.

### **Weaknesses**

*  While highly competitive, the method does not completely close the performance gap with dense model training. For instance, reducing the density level to 0.25 results in a marginal validation perplexity increase of approximately 1 point, and this degradation becomes slightly more pronounced at lower density levels[cite: 768].
*  The authors acknowledge that while SMET drastically reduces memory consumption, achieving practical computational speedups during both training and inference is still heavily bottlenecked by current hardware support for unstructured sparsity patterns.
* The current analysis, experiments, and theoretical framework are focused on Adam-based optimizers. The paper leaves it unexplored whether similar cold-start instabilities occur under other adaptive optimization schemes e.g. muon.

---

> ### Author Rebuttal · Authors · 2026-03-30
>
> Thank you very much for your thoughtful questions.
>
> > w1: While highly competitive, the method does not completely close the performance gap with dense model training.... this degradation becomes slightly more pronounced at lower density levels.
>
> We agree that, although SMET substantially narrows the gap to dense training, it does not completely eliminate it, especially at very low density levels. However, these regimes also provide much larger memory savings, which is a main goal of our method (Figure 3 in the paper).
>
> Importantly, **the gap shrinks as model size increases**. In our additional scaling experiments (Table 1), at d=0.125 the gap drops from 2.97 on LLaMA2-350M to **1.28 on 2B**. This suggests that SMET has even greater potential at larger model scales, where sparse training appears to become more favorable.
>
> We will make this trade-off and scaling trend more explicit in the revised version.
>
> Table 1. Perplexity($\downarrow$) on models trained for 2.6B tokens under different model sizes.
> |Model|LLaMA2-350m | LLaMA2-1B | LLaMA2-2B
> |-|-|-|-|
> |Dense| 21.38 |20.61 | 19.25
> |d=0.125| 24.35(+2.97) |22.34(+1.73) |20.53(+1.28)
>
>
> > w2: Achieving practical computational speedups during both training and inference is still heavily bottlenecked ...
>
> We agree that the computational benefits of unstructured sparsity are not yet fully realized on current hardware infrastructures.
>
> Recent developments in sparse hardware and system design are increasingly moving toward better utilization of sparsity. For example, NVIDIA’s A100 GPU supports **2:4 sparsity** as described in [1], and other innovations are making strides toward **efficient sparse implementations** [2,3]. Simultaneously, **software libraries** such as [4] are emerging to enable truly sparse network implementations. We believe these advancements are paving the way for future deep neural networks to achieve greater efficiency in terms of computation.
>
> Importantly, SMET is not limited to fully unstructured sparsity. It can be **naturally extended to block-wise sparsity**, which is more compatible with current hardware. Moreover, because SMET trains sparse models throughout training rather than relying on post-training pruning, the final model is already sparse and can be directly used for sparse inference, **where we already observe practical speedup, 2× speedup at 90% sparsity  (Table 2)**.
>
> [1] Zhou, Aojun, et al. Learning N: M Fine-grained Structured Sparse Neural Networks From Scratch. ICLR 2021.
>
> [2] Chen, Yu-Hsin, et al. Eyeriss v2: A flexible accelerator for emerging deep neural networks on mobile devices. 2019.
>
> [3] Ashby, Mike, et al. Exploiting unstructured sparsity on next-generation datacenter hardware. (2019).
>
> [4] Liu, Shiwei, et al. Sparse evolutionary deep learning with over one million artificial neurons on commodity hardware. (2021).
>
> Table 2. Inference speedup of LLaMA2-1B under different density levels.
> |Sparsity|Latency(ms)|Throughput (TFLOPs)|
> |-|-|-|
> |0.00|256.94|236.41|
> |0.50|219.72|276.45|
> |0.70|182.07|333.63|
> |0.80|160.00|379.65|
> |0.90|133.57|454.77|
> |0.95|120.00|506.18|
>
>
> > w3: The current analysis, experiments, and theoretical framework are focused on Adam-based optimizers….
>
> We agree that the current analysis is primarily focused on Adam-based optimizers, which remain the standard choice for large-scale model training. Based on our finding, the cold-start mechanism we study is directly tied to **moment initialization and bias-correction** in Adam.
>
> To examine generality, we also tested Muon on LLaMA2-350M with 2.6B training tokens. We find that **SMET still improves sparse training under Muon**, but the gain is smaller than in Adam. This is consistent with the fact that Muon has different optimizer dynamics and does not exhibit the same bias-correction behavior.
>
> Therefore, our current interpretation is that the regrowth instability is **not entirely unique to Adam**, but its severity and the effectiveness of our stabilization strategy do depend on the optimizer.
>
> Table 3. Perplexity($\downarrow$) on LLaMA2-350M trained for 2.6B tokens.
> |Model|dense | d=0.5(SET) | d=0.5(Ours)
> |-|-|-|-|
> |Muon| 18.82 | 20.73 | 19.71
> |Adam|21.38 | 24.71 | 21.61
>
>
> > w4: Does the performance gap with dense training widen or narrow when scaling beyond ...?
>
> We thank the reviewer for this important question. Our additional scaling experiments show that the **sparse-dense gap narrows as model size increases (Table 1)**. For the same density, larger models stay closer to their dense counterparts than smaller models do.
>
> This provides encouraging evidence that SMET remains effective beyond the 1B regime and is promising for **scaling sparse training to larger models**. We will make this trend more explicit in the revised paper.
>
> Thank you for your feedback on our response. Please let us know if you still have any additional questions or would like further clarification.

---

> > ### Author Rebuttal · Reviewer_kp99 · 2026-03-31
> >
> > I thank author for their detailed response. I do not have further question and decide to maintain my score.

---

> > > ### Author Response · Authors · 2026-04-05
> > >
> > > We are pleased to have successfully addressed your concerns. We sincerely appreciate your thorough review and valuable feedback, which helped improve the overall quality of our work. Thank you for your time and effort throughout this review process.
> > >
> > > Best regards,
> > >
> > > Authors

---

### Decision · Program_Chairs · 2026-04-30

**Decision:**

Accept (regular)

**Comment:**

This paper studies instability in dynamic sparse training for LLM pre-training and proposes SMET to fix it. The paper’s main contribution is to identify the cold-start issue after topology updates, analyze training instability, and introduce a practical solution.

The paper’s strengths are clear practical motivation, strong empirical performance. Reviewers found the method simple and effective, and the rebuttal clarified several important points, including the role of SMET as a wrapper on top of DST methods, its compatibility with other approaches, and evidence on scaling, memory savings, and broader applicability.

The main concerns were about the limited theory, the heuristic nature of some design choices, and the original presentation and experimental scope. In particular, reviewers noted that the theoretical analysis is local rather than a full convergence result, and that some claims needed clearer qualification. That said, these concerns were substantially addressed during rebuttal, and the reviewers ultimately converged on support for acceptance. After checking the empirical setup, I also have one concern that the training are conducted under bf16 precision, which is uncommon for standard LLM pretraining. Mixed precision is typically used, and they can have drastically different behaviors. Therefore, I recommend the author should clearly state this in the revised version or add mixed-precision experiments.

Nevertheless, I recommend acceptance since its practical contribution outweighs the potential concerns.